# Identification of Key Genes Affecting Flavor Formation in Beijing-You Chicken Meat by Transcriptome and Metabolome Analyses

**DOI:** 10.3390/foods12051025

**Published:** 2023-02-28

**Authors:** Kai Gai, Yu Ge, Dapeng Liu, He Zhang, Bailin Cong, Shihao Guo, Yizheng Liu, Kai Xing, Xiaolong Qi, Xiangguo Wang, Longfei Xiao, Cheng Long, Yong Guo, Xihui Sheng

**Affiliations:** 1Animal Science and Technology College, Beijing University of Agriculture, Beijing 102206, China; 2Institute of Animal Science, Chinese Academy of Agricultural Sciences, Beijing 100193, China

**Keywords:** meat flavor, Beijing-You chicken, transcriptome, metabolome, regulatory network

## Abstract

The flavor of chicken meat is influenced by muscle metabolites and regulatory genes and varies with age. In this study, the metabolomic and transcriptomic data of breast muscle at four developmental stages (days 1, 56, 98, and 120) of Beijing-You chickens (BJYs) were integrated and 310 significantly changed metabolites (SCMs) and 7,225 differentially expressed genes (DEGs) were identified. A Kyoto Encyclopedia of Genes and Genomes (KEGG) enrichment analysis showed that SCMs and DEGs were enriched in amino acid, lipid, and inosine monophosphate (IMP) metabolism pathways. Furthermore, genes highly associated with flavor amino acids, lipids, and IMP were identified by a weighted gene co-expression network analysis (WGCNA), including cystathionine β-synthase (*CBS*), glycine amidinotransferase (*GATM*), glutamate decarboxylase 2 (*GAD2*), patatin-like phospholipasedomain containing 6 (*PNPLA6*), low-specificity L-threonine aldolase (*ItaE*), and adenylate monophosphate deaminase 1 (*AMPD1*) genes. A regulatory network related to the accumulation of key flavor components was constructed. In conclusion, this study provides new perspectives regarding the regulatory mechanisms of flavor metabolites in chicken meat during development.

## 1. Introduction

The growth efficiency of chickens has increased rapidly as large-scale poultry production has increased, but the quality and flavor of chicken have decreased significantly. The meat color, odor, and taste of chicken are the decisive factors that affect consumption [1]. Among them, chicken flavor (taste and odor) is an important index to measure quality [2]. Therefore, improving meat flavor has become an important research topic in broiler breeding. Meat flavor is attributed to a variety of volatile compounds formed during cooking by reactions between low molecular weight water-soluble compounds and lipids [3]. However, flavor phenotypes are difficult to quantify and are strongly influenced by environmental factors [4], which makes it difficult to improve chicken flavor.

Lipids and water-soluble components are the main flavor precursors of meat. Water-soluble components include free sugars, sugar phosphates, nucleotide-bound sugars, free amino acids, peptides, nucleotides, and sulfur-containing compounds [5]. The umami flavor of chicken meat is primarily derived from water-soluble precursors substances such as inosine monophosphate (IMP) and glutamate [6,7]. Lipids are the fat-soluble substances found in meat. The main components of intramuscular fat (IMF) and subcutaneous fat are triglycerides and phospholipids, which contain large amounts of unsaturated fatty acids such as linolenic acid, oleic acid, and arachidonic acid (ARA). It has been demonstrated that chickens with a high ARA content have better sensory quality [8,9], and oleic acid is also associated with the taste of meat [10]. According to research by Mottram et al., phospholipids are important precursors for meat flavor, whereas triglycerides have little impact on flavor [11]. Studies have shown that adenylosuccinate lyase (*ADSL)* is a key gene in regulating the IMP synthetic pathway, and fatty acid-binding proteins (*FABPs*) and peroxisome proliferator-activated receptor-γ (*PPARγ*) are important genes that regulate lipid transport and metabolism. However, the key metabolites and molecular mechanisms that influence chicken flavor have not been fully elucidated.

The transcriptome may present different gene expression states under different conditions. Furthermore, metabolomic analysis reveals changes in metabolites caused by gene regulation. Many researchers have investigated the regulatory process of IMF formation in livestock and poultry meat through a joint analysis of the transcriptome and metabolome [12,13], which has improved our understanding of the regulatory mechanisms involved in meat flavor.

The Beijing-You chicken (BJY) is a distinctive indigenous breed that is well known for its meat quality and flavor in China. The free amino acids in its muscles are significantly higher than those of other breeds [2], and it is also rich in essential fatty acids and phospholipids [14]. In this study, we performed joint analyses between the transcriptome and metabolome to investigate the metabolic dynamics of flavor precursors in breast muscle samples of BJYs obtained on days 1, 56, 98, and 120, and identified key metabolites and genes that can affect meat flavor. Our results provide a theoretical foundation for the molecular mechanisms underlying chicken flavor and utilization of the germplasm resources of BJYs.

## 2. Materials and Methods

### 2.1. Ethics Approval

Animal welfare practices and experimental procedures were performed in accordance with the Guide for the Care and Use of Laboratory Animals (Ministry of Science and Technology of China, 2006). All procedures were approved by the Animal Ethics Committee of the Beijing University of Agriculture.

### 2.2. Animals

The BJYs used in this experiment were obtained from a farm in a suburb of Beijing. A total of 100 1-day-old chickens were randomly selected and raised under identical standard management conditions with free access to water. The ingredients and composition of the BJY diets at different feeding stages are shown in Appendix A. The immune procedures at different developmental stages are shown in Appendix A. Ten healthy chickens of similar body weight were chosen at 1 (birth stage), 56 (rapid growth stage), 98 (stage with high deposition of IMF), and 120 days of age (marketing stage), respectively, and were slaughtered by a conventional neck cut, bled, and plucked. Breast muscle without skin was then collected from similar sampling sites and stored in a refrigerator at −80 ℃. The body weight and breast muscle weight at the four developmental stages of BJYs are shown in Appendix A.

### 2.3. RNA Extraction and Sequencing

Total RNA was extracted from 40 frozen breast muscle samples using TRIzol reagent (Invitrogen, Carlsbad, CA, USA) according to the manufacturer’s instructions. An ARNA 6000 Nano LabChip Kit for Bioanalyzer 2100 (Agilent, Santa Clara, CA, USA) was used to determine the purity and concentration of RNA. Forty cDNA libraries were created by reverse transcription using an mRNA-Seq Sample Preparation Kit (Illumina, San Diego, CA, USA). A HiSeq 2500 instrument was used for paired-end sequencing (Illumina). Raw data (raw reads) in fastq format were filtered to ensure quality. Reads with adapter sequences and low quality were eliminated and all reads containing A bases and reads with N ratios >10 % were eliminated to ensure quality. Using Hisat2 with the default settings, clean paired-end reads were aligned to the chicken reference genome (version: GCF_016699485.2). Stringtie (v2.1.5, http://ccb.jhu.edu/software/stringtie/, accessed on 3 September 2021) was used to assemble transcripts. Gene expression levels were estimated using counts per million (CPM).

### 2.4. Analysis of Gene Expression Data

The differentially expressed genes (DEGs) between the two stages were calculated using the edger R package, which was defined as genes with a false discovery rate (FDR) ≤ 0.05 and a |log2(fold change)| ≥ 1. The DEGs were analyzed, clustered, and visually represented using Short Time-series Expression Miner (STEM) software (Pittsburgh, PA, USA, v1.3.11). The minimum and maximum numbers of model profiles were 2 and 45, respectively. We standardized the data using Log_2_ (CPM), and the screening interval for a valid trend was *p* < 0.05. A hierarchical cluster analysis of the DEGs was performed using the ggplot2 package. Gene Ontology (GO) and Kyoto Encyclopedia of Genes and Genomes (KEGG) pathway enrichment analyses were performed using R package clusterProfiler (Guangdong, China, v4.0.5).

### 2.5. Metabolite Extraction and LC-MS/MS Conditions

#### 2.5.1. Metabolite Extraction

A total of 40 frozen samples from four developmental stages were defrosted. Following homogenization, the samples were centrifuged at 12,000 rpm for 10 min at 4 °C and the supernatant was collected in a tube for the LC-MS/MS analysis.

#### 2.5.2. LC-MS/MS Conditions

The sample extracts were analyzed using an LC-MS/MS system (UFLC, shimadzu UFLC SHIMADZU CBM30A, available at https://www.shimadzu.com) (accessed on 3 September 2021) and chromatographic separations were performed. In reverse phase separation, a Waters ACQUITY UPLC HSS T3 C18 (1.8 m, 2.1 mm × 100 mm) was employed. The HPLC conditions were as follows: solvent system, water (0.04 % acetic acid): acetonitrile (0.04% acetic acid); gradient program of 95:5 *V/V* at 0 min, 5:95 *V/V* at 11.0 min, 95:5 *V/V* at 12.1 min, and 95:5 *V/V* at 14.0 min; flow rate, 0.40 mL.min^−1^; temperature, 40 °C; injection volume, 2 μL.

A QTRAP^®^ 6500+ LC-MS/MS System (https://sciex.com/) (accessed on 3 September 2021), equipped with an electrospray ionization (ESI) Turbo Ion-Spray interface and operating in positive and negative ion mode was used. The operating parameters for the ESI source were as follows: the ion spray (IS) voltage was 5500 V (positive) and −4500 V (negative) and the turbo spray source temperature was 500 °C. The ion source gas I (GSI), gas II (GSII), and curtain gas (CUR) pressures were 55, 60, and 25.0 psi, respectively. The level of the collision gas (CAD) was high. Instrument calibration and mass calibration were performed with solutions of 10 and 100 mol/L polypropylene glycol in the QQQ and LIT modes, respectively. A specific set of multiple reaction monitoring (MRM) transitions was observed based on the metabolites eluted each time.

### 2.6. Qualitative, Quantitative and Statistical Analysis of Metabolites

Metabolites were differentiated by contrasting the m/z values of precursor ions, retention times, and fragmentation patterns with the standards in a database created by MetWare Biotechnology Co., Ltd. (Wuhan, China). An MRM pattern was used for quantitative detection. The following screening thresholds were used to identify significantly changed metabolites (SCMs): |log2(fold change)| ≥ 1 and variable importance in projection (VIP) ≥ 1. Principal component analysis (PCA) was carried out using SIMCA14.1 software. MetaboAnalyst 5.0 (https://www.metaboanalyst.ca/) (accessed on 3 September 2021) was used to analyze metabolic pathways of the SCMs.

### 2.7. Integrative Analysis of Metabolomic and Transcriptomic Datasets

Co-expression network modules of all DEGs were constructed using the WGCNA R package (v1.70-3) and analyzed in combination with metabolites. The automatic network creation function (blockwiseModules) with the default parameters was used to obtain co-expression modules. The parameters were minModuleSize = 30, mergeCutHeight = 0.25, and soft threshold power = 10. Screening was carried out utilizing a gene significance |GS| ≥ 0.7 and a module membership |MM| ≥ 0.7 in order to better investigate interactions between genes in the modules. The link between genes and important metabolites was determined using Pearson correlation analysis. Interaction networks were constructed using Cytoscape (https://cytoscape.org) (accessed on 3 September 2021).

KEGG pathways that were jointly enriched by all SCMs and DEGs were analyzed. The ‘cor’ package in R software (www.r-project.org) (accessed on 3 September 2021) was used to calculate Pearson correlation coefficients between the SCMs and DEGs through pairwise comparisons.

## 3. Results

### 3.1. Transcriptome and DEGs Analysis

RNA-seq analyses were conducted to examine gene expression profiles of breast muscles in BJYs at different developmental stages. A total of 7225 DEGs were identified in this study: 6,521 at day 1 vs. 56, 819 at day 56 vs. 98, and 747 at day 98 vs. 120 (Figure 1A). The results of cluster analysis revealed differential expression of genes at different stages (Figure 1B). To identify genes that play key roles in breast muscle development, 82 critical genes were discovered at the intersection generated from a Venn diagram of DEGs (Figure 1C and Appendix A).

### 3.2. GO and KEGG Enrichment Analysis of DEGs

We investigated the functions of 7,225 DEGs using GO analysis. Biological processes contained 268 significant terms (*p* < 0.05), and the top five terms in this category were cellular developmental process, cell differentiation, tissue development, cell proliferation, and regulation of cell differentiation. Molecular functions involved 18 significant terms, including cytoskeletal protein binding, lipid binding, tubulin binding, and sulfur compound binding (Figure 2A and Appendix A). The KEGG enrichment analysis identified 26 key KEGG pathway terms, including multiple pathways involved in muscle growth, including the Wnt, MAPK, and PPAR signaling pathways (Figure 2B and Appendix A).

The functions of 82 intersecting genes in a Venn diagram analysis were analyzed to determine their involvement in regulatory mechanisms surrounding meat flavor. KEGG pathway analysis showed that multiple genes were enriched in pathways associated with muscle development, including ECM–receptor interaction, PPAR signaling pathway, and fatty acid biosynthesis and degradation, in which secreted phosphoprotein 1 (*SPP1*), thrombospondin 1 (*THBS1*), thrombospondin 2 (*THBS2*), and acyl-CoA synthetase long chain family member 4 (*ACSL4*) were associated with the biological pathways of IMF deposition (Table 1).

### 3.3. Time Series Analysis of DEGs

STEM analysis revealed that the expression patterns of 7225 DEGs in breast muscle during BJY development were enriched into forty-five profiles, of which nine profiles appeared significant, including six up-regulated and three down-regulated profiles. A total of 2931 DEGs exhibited an up-regulation trend and were significantly enriched in profiles 5, 10, 36, 38, 42, and 44, whereas 2830 DEGs showed a down-regulation trend and were significantly enriched in profiles 0, 2, and 4 (Figure 2C). The genes enriched in profiles 2 and 38 were the most abundant, and KEGG enrichment analysis revealed that genes in profile 2 were primarily enriched in the Wnt signaling and purine metabolism pathways, whereas the genes in profile 38 were primarily enriched in cytokine–cytokine receptor interaction and cell adhesion molecular pathways (Appendix A).

### 3.4. Metabolomic Data and SCMs Analysis

We further analyzed the metabolic alterations in the breast muscle of BJYs at four developmental stages using an LC-MS/MS approach. In total, 578 compounds were identified and classified into 33 classes; primarily organic acids, carbohydrates, lipids, nucleotides and their derivatives, vitamins, and amino acid derivatives. A PCA of the metabolic data of the four developmental stages indicated a good correlation between replicates, and SCMs from day 1 were clearly distinct from those of other stages (Figure 3A). We identified 310 SCMs using the criteria of a VIP ≥ 1 and a |log2(fold change)| ≥ 1. At day 1 vs. 56, we found 294 SCMs, of which 221 and 73 were down- and up-accumulated metabolites, respectively. At day 56 vs. 98, a total of 19 SCMs, including 13 down- and 6 up-accumulated metabolites were detected. A total of 21 SCMs were identified at day 98 vs. 120, including 12 down- and 9 up-accumulated metabolites (Figure 3B). KEGG enrichment analysis indicated that the majority of SCMs were involved in amino acid metabolism, such as arginine biosynthesis; arginine and proline metabolism; purine metabolism; glycine, serine, and threonine metabolism; and alanine, aspartate, and glutamate metabolism (Figure 3C and Appendix A).

### 3.5. Joint Analysis of Transcriptomic and Metabolomic Data

To better understand the gene regulation mechanism of metabolites during BJY development, nine co-expression modules of DEGs were identified using WGCNA (Figure 4A).

DEGs of BJYs from day 1 vs. 56 were mainly gathered in the blue module, whereas DEGs from day 56 vs. 98 and from day 98 vs. 120 were mostly located in the cyan module. According to the heatmap of module–trait relationships (Figure 4B), the accumulation of transcripts of the blue module correlated with amino acids and lipid metabolites, including serine, glycine, cysteine, threonine, glutamate, and lysophosphatidylcholine (LPC), which are the main flavor compounds of BJYs. The accumulation of transcripts in the cyan module was correlated with flavor-associated metabolites such as creatine and IMP. These results indicated that the DEGs in these modules were mainly associated with flavor formation during the development of BJY breast muscle.

### 3.6. Generation of Flavor Metabolic Regulatory Networks

To further explore the relationship between DEGs and SCMs, DEGs in the blue and cyan modules with a gene significance, |GS|, of ≥0.7, and a module membership, |MM|, of ≥0.7 were used to analyze interactions with SCMs. We identified 11 genes in the blue module involved in the amino acid metabolic pathway, including cystathionine β-synthase (*CBS*), glutamate decarboxylases (*GAD1* and *GAD2*), glycine amidinotransferase (*GATM*), glycine decarboxylase (*GLDC*), glycine N-methyltransferase (*GNMT*), low-specificity L-threonine aldolase 2 (*ItaE*), cysteine lyase (*CYLY*), sarcosine dehydrogenase (*SARDH*), cysteine dioxygenase (*CDO1*), and serine racemase (*SSR*) genes, the expressions of which were highly correlated with the accumulation of glutamate, serine, glycine, threonine, and cysteine (Figure 4C).

LPC plays a vital role in the formation of meat flavor. We identified key regulatory genes involved in the LPC biosynthesis pathway in the blue module, including acylglycerol phosphate acyltransferases (*AGPAT2* and *AGPAT4*), diacylglycerol kinases (*DGKB*, *DGKH*, and *DGKI*), glycerol-3-phosphate acyltransferases (*GPAM* and *GPAT2*), lecithin cholesterol acyltransferase (*LCAT*), phospholipid phosphatases (*PPAP2B* and *PPAPDC1B*), acetylcholinesterase (*ACHE*), ethanolamine kinase (*ETNK2*), patatin-like phospholipase (*PNPLA6*), and phospholipase (*PLA2G4EL1* and *PLA2G4A*) genes (Figure 4D).

IMP is an important flavor substance in chickens. Seventeen genes in the cyan module were identified as good candidates encoding key genes in the IMP biosynthesis pathway, including adenylate monophosphate deaminases (*AMPD1* and *AMPD3*), adenylosuccinate synthetase (*Adssl1*), ectonucleoside triphosphate diphosphohydrolases (*ENTPD5* and *ENTPD6*), phosphodiesterases (*PDE1B*, *PDE4B*, *PDE8B*, *PDE10A*, and *PDE4D*), adenylate kinases (*AK1* and *AK3*), 5′,3′-nucleotidase (*NT5C1A* and *NT5M*), one ectonucleotide pyrophosphatase (*ENPP3*), nucleoside diphosphate kinase (*NME3*), and phosphoribosylglycinamide formyltransferase (*GART*) genes (Figure 4E).

### 3.7. Integrated Analysis of Flavor Formation during BJY Development

A KEGG enrichment analysis of DEGs and SCMs revealed 46 co-enriched pathways (Figure 5A). The pathways associated with amino acids, lipids, and IMP were highlighted in this analysis, and these processes were combined in the related network (Figure 5B). We observed that the network involved flavor metabolites including glycine, serine, cysteine, threonine, and glutamate. The content of these amino acids was higher during the early development of BJYs. In the glycerophospholipid metabolism pathway, the LPC content increased from day 1 to 56 and decreased gradually with continuing growth, and was the key compound that led to fat deposition in the late stage of BJY maturation. IMP content in the purine metabolism pathway significantly increased with the age of BJYs. There was a significant correlation between the regulatory genes and flavor metabolites in the network (Figure 5C). For example, the correlation of *CBS* with cysteine and serine was 0.93 and 0.9, respectively; glycine with *GATM* was 0.88; IMP with *AMPD1* was 0.65; and LPC with *PNPLA6* was 0.54 (*p* < 0.05).

## 4. Discussion

Elucidating the regulatory mechanisms involved in the synthesis and accumulation of flavor compounds is essential for improving chicken meat quality. As illustrated in this study, 310 SCMs and 7,225 DEGs were identified at four developmental stages of BJYs. The functions of DEGs and SCMs were analyzed by KEGG enrichment, and the results showed that they were co-enriched in glycerophospholipid metabolism; glycine, serine, and threonine metabolism; purine metabolism; and alanine, aspartate, and glutamate metabolism pathways. To further understand the molecular mechanisms involved in meat flavor formation during the development of BJYs, we identified genes significantly related to flavor metabolites using WGCNA and established a regulatory network related to the accumulation of key flavor components. *CBS*, *GATM*, *GAD2*, *PNPA6*, *ItaE*, *AMPD1*, glycine, serine, cysteine, and threonine have been identified as important genes and metabolites that affect flavor formation. This study not only helps to define the regulatory networks of specific flavor compounds in BJY meat but also provides a theoretical foundation for the improvement of meat flavor in broiler breeds.

Amino acids are crucial components in meat flavor [15]. The degradation of peptides and amino acids in meat improves its sensory properties and taste [3]. Glutamate, succinic acid, and IMP are umami amino acids, whereas glycine, threonine, alanine, and serine are sweet amino acids [16]. Cysteine is the cause of sulfur-containing flavors in meat [17], as it can produce meat flavors during heating [18]. However, the precise regulatory pathways of amino acid-derived flavors in BJYs remain unknown. We identified genes related to amino acid metabolism by WGCNA, including *CDO1*, *CYLY*, *CBS*, *GAD*, *GATM*, and *ItaE*. *CDO1*, *CYLY,* and *CBS* are considered critical genes involved in cysteine metabolism. *CDO1* regulates cysteine concentrations in mice by participating in cysteine degradation [19]. Cysteine is the only major substrate of *CYLY* [20]. *CBS* catalyzes the formation of cysteine by the condensation of serine and homocysteine with water [21]. The absence of *CBS* increases the risk of elevated plasma homocysteine levels and severe growth retardation [22,23]. In addition, we demonstrated a high correlation between *CBS* and cysteine (*r* = 0.93). The consistent expression trends of *CBS* and cysteine indicated that *CBS* may play an important role in the growth and flavor formation of BJYs. *GAD* transforms glutamate to γ-aminobutyric acid by decarboxylation, which is then converted to succinate [24]. This was also confirmed by the higher association between *GAD2* and glutamate in this study (*r = 0.72*). *GATM* encodes a mitochondrial enzyme that belongs to the amidinotransferase family and is thought to be a key gene in the process of creatine metabolism [25]. In this study, the creatine and glycine expression trends were opposite. A significant association between glycine and *GATM* supported speculation that *GATM* expression controls creatine metabolism and leads to glycine accumulation. Under physiological conditions, *ItaE* performs threonine catabolism and glycine synthesis by catalyzing the cleavage of threonine into glycine and acetaldehyde [26]. Similarly, our results showed that decreased glycine levels were associated with a decreased *ItaE* expression. The coefficients of correlation of *ItaE* with threonine and glycine were 0.48 and 0.59, respectively. Finally, we speculated that *CBS*, *GATM*, *GAD2*, and *ItaE* were key genes involved in amino acid-derived flavor formation in breast muscle during BJY development.

LPC is obtained by loss of a fatty acid group from lecithin. In vitro studies have confirmed that the ability of LPC to emulsify fat is 4–5 times higher than that of common oil. Our results showed that LPC expression levels decreased with the aging of BJYs, which may be related to fat deposition in late developmental stages of BJYs. We identified a total of 15 DEGs significantly associated with LPC in the blue module. The PNPLA6 enzyme can react with a variety of substrates, including retinol esters, triacylglycerols, and phospholipids [27]. However, it can preferentially hydrolyze phosphatidylcholine (PC) and LPC [28]. In both *Drosophila* and mice, *PNPLA6* gene deletion leads to increased lipid deposition, motor impairments, and neurodegeneration [29]. In this study, we identified the transcription level of *PNPLA6* was significantly positively correlated with LPC (*r* = 0.54). We speculated that the decrease in LPC content under the regulation of *PNPLA6* led to the accumulation of large amounts of fat during the late development of BJYs. The AGPAT enzyme catalyzes the conversion of lysophosphatidic acid (LPA) to phosphatidic acid (PA) [30]. PA is a substrate for the synthesis of other polar phospholipids (PL) such as PC, phosphatidylserine (PS), and phosphatidylethanolamine (PE) [31]. *AGPAT2* encodes a member of the 1-acylglycerol-3-phosphate O-acyltransferase family, which is involved in the conversion of lysophosphatidic acid to phosphatidic acid during the second step of phospholipid biosynthesis. The GPAT enzyme converts glycerol-3-P to lysolecithin, which is subsequently acylated to PA [32]. In addition, PA contains two fatty acids in its glycerol backbone at the sn-1 and sn-2 positions. Although GPAT determines fatty acid specificity at the sn-1 (carbon 1) position, AGPAT enzyme esterification causes variability at the sn-2 (carbon 2) position [33,34]. As a result, these two enzymes can produce a wide range of PA species with varying fatty acids at the two carbon positions. Based on our results, we believe that these genes are involved in fat metabolism, and the regulatory effects need to be confirmed by further research

The Maillard reaction refers to the polymerization, condensation, and other reactions involving compounds containing free amino groups and reducing sugars or carbonyl compounds at normal atmospheric temperatures or under heating. IMP degradation-forming ribose participates in the Maillard reaction, which is also an important reaction in flavor formation. Numerous studies have shown that IMP is one of the most important flavor components in meat [35,36]. The results of this study showed that IMP content was the highest at the age of 56 days, and then decreased gradually with the increase in age, which was consistent with Katemala et al.’s study on Korat hybrid chickens [37]. Previous studies involving the formation of IMP have confirmed that AMPD1 enzyme catalyzes the irreversible hydrolysis of adenosine 5′-monophosphate (AMP) to IMP and ammonia [38]. Our data verified a significantly high correlation between *AMPD1* and IMP (*r* = 0.65). Therefore, this gene was selected as a key gene that controls IMP biosynthesis by regulating AMP metabolism. In addition, we also identified DEGs involved in the IMP metabolic pathway. *GART* and *ADSSL1* genes were able to increase the synthesis of IMP by promoting the purine de novo synthesis. Purine de novo synthesis was divided into 10 steps, in which GART was involved in Steps 2, 3, and 5, and ADSLL1 was involved in the Step 8 reaction in this pathway [39]. AMPD3 and NT5C are involved in the degradation of purine to accelerate the de novo synthesis of purine [40]. We found that the expression levels of these genes were significantly correlated with IMP. These results showed that these genes were related to IMP metabolism in the breast muscle of BJYs.

## 5. Conclusions

In conclusion, we identified four crucial metabolic pathways involved in flavor formation in BJY breast muscle by combining transcriptomic and metabolomic data, including glycerophospholipid metabolism; glycine, serine, and threonine metabolism; alanine, aspartate, and glutamate metabolism; and purine metabolism pathways. We also identified a number of important metabolites and regulatory genes affecting meat quality and flavor involving these pathways, including glycine, serine, glutamate, threonine, LPC, IMP, *CBS*, *GATM*, *GAD2*, *PNPLA6*, *ItaE*, and *AMPD1*. However, the genetic mechanisms of chicken quality and flavor and the molecular function of these key genes and metabolites need to be further verified. In brief, our research provides not only new perspectives regarding the regulatory metabolism of chicken meat flavor during development, but also a theoretical foundation for the utilization of BJYs and genetic improvement in broiler meat quality and flavor.

## Figures and Tables

**Figure 1 foods-12-01025-f001:**
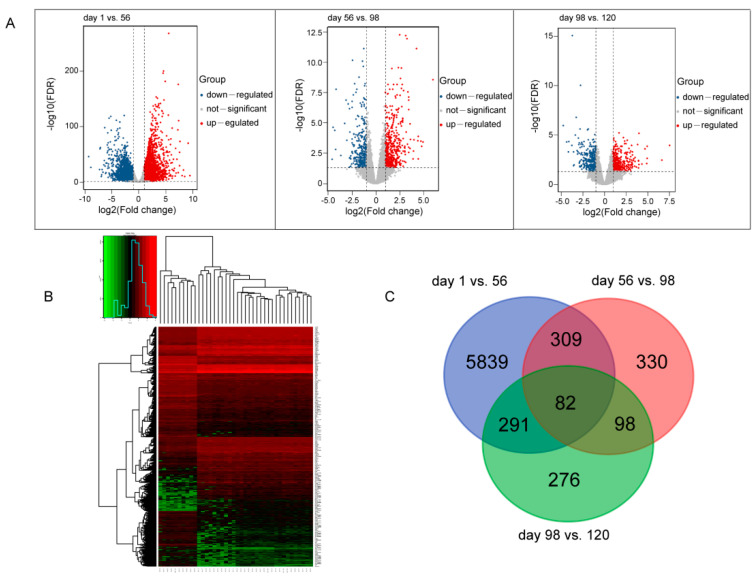
Differential expression analysis of mRNAs of breast muscle at four developmental stages of Beijing-You chicken (BJY). (**A**) Differentially expressed genes (DEGs) between different stages. (**B**) Heatmap of DEGs by cluster analysis. (**C**) Venn diagram plot of DEGs.

**Figure 2 foods-12-01025-f002:**
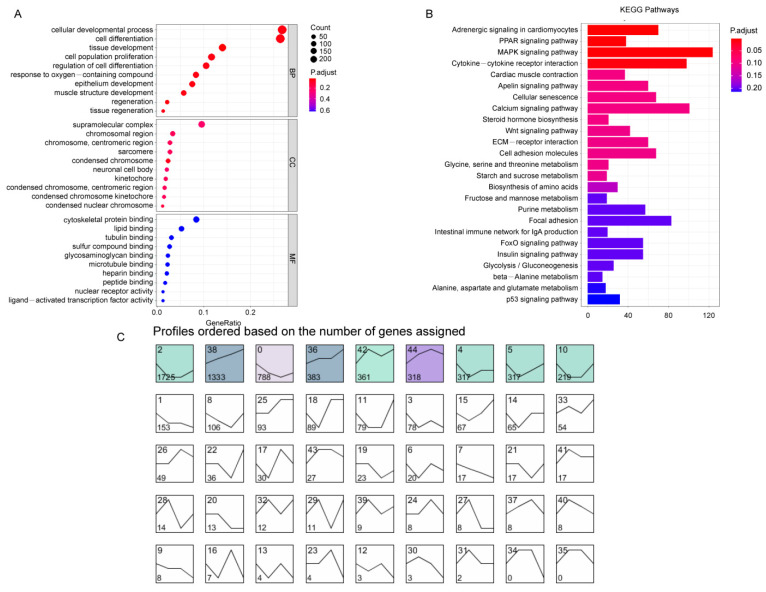
Functional analysis and sample time series analysis of differentially expressed genes (DEGs) of breast muscle at four developmental stages of Beijing-You chicken (BJY). (**A**) The top 10 significant GO enrichment terms. (**B**) The top 25 significant terms by KEGG enrichment analysis. (**C**) Short time-series expression miner (STEM) clustering of DEGs, the color indicates significant difference (*p* < 0.05) and gray indicates no significant difference (*p* > 0.05).

**Figure 3 foods-12-01025-f003:**
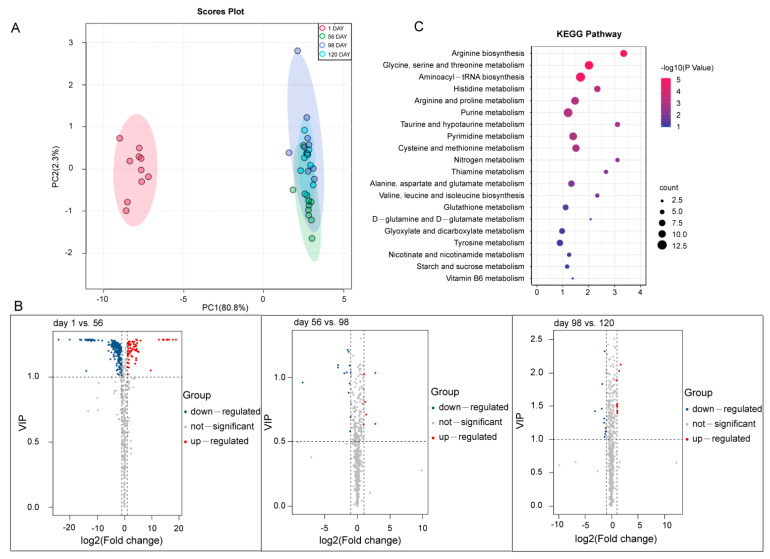
Analysis of metabolites of breast muscle at four developmental stages of Beijing-You chicken (BJY). (**A**) Principal component analysis (PCA) of the identified metabolites. The X axis represents PC1 and the Y axis represents PC2. Each sample had ten biological replicates. (**B**) Significantly changed metabolites (SCMs) between different stages. (**C**) The top 25 significant terms of SCMs by KEGG analysis.

**Figure 4 foods-12-01025-f004:**
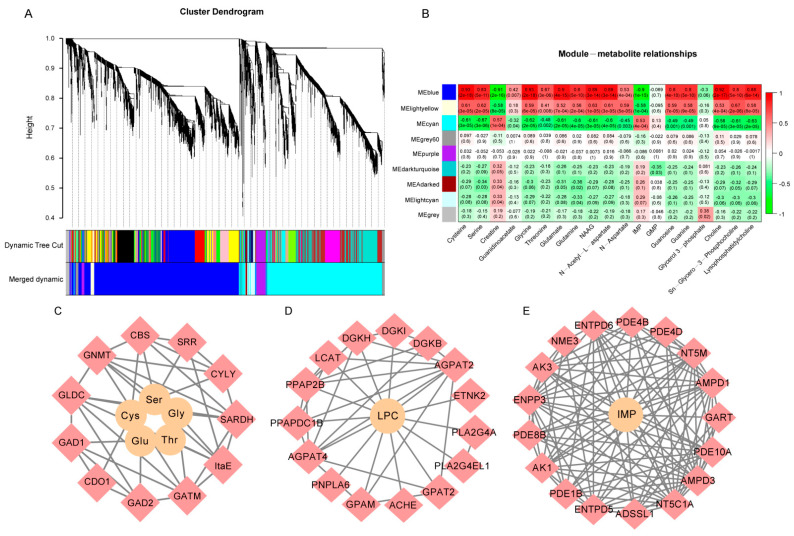
Joint analysis of significantly changed metabolites (SCMs) and differentially expressed genes (DEGs) of breast muscle at four developmental stages of Beijing-You chicken (BJY). (**A**) Nine clustering modules with different expression trends of DEGs obtained by overweighted gene co-expression network analysis (WGCNA). The dendrogram shows DEG co-expression clusters. Different colors indicate DEG co-expression modules. (**B**) Heatmap showing module–metabolite relationships. Each row represents a different modules obtained from the WGCNA analysis. Each column represents a metabolite. Red indicates that there was a positive correlation between this cluster and the metabolite, and green indicates a negative correlation. The numbers in the module indicate the corresponding *p*-value and correlation coefficient. (**C**) Interaction network of amino acids and DEGs by cytoscape. Yellow circles represent amino acids. Pink diamonds represent DEGs involved in amino acids metabolism. (**D**) Interaction network of lysophosphatidylcholine (LPC) and DEGs. Yellow circles represent LPC. Pink diamonds represent DEGs involved in LPC biosynthesis. (**E**) Interaction network of inosine monophosphate (IMP) and DEGs. Yellow circles represent IMP. Pink diamonds represent DEGs involved in IMP biosynthesis.

**Figure 5 foods-12-01025-f005:**
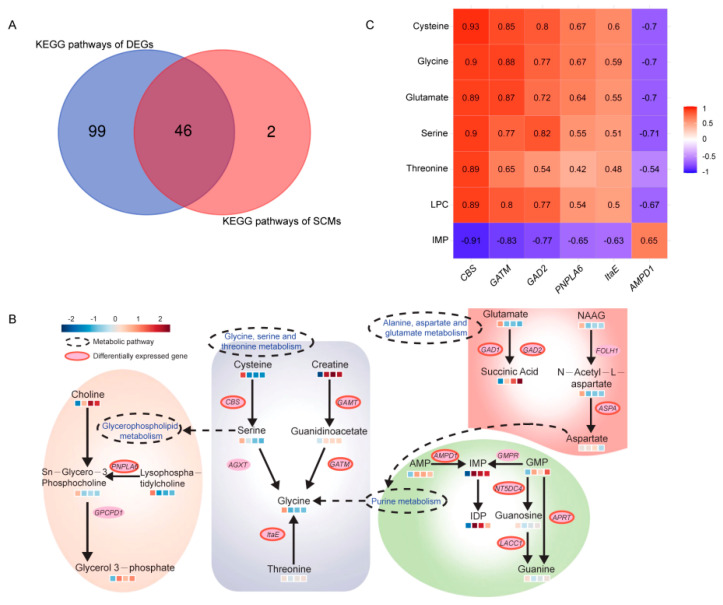
Identification of key metabolic pathways and genes related to the meat flavor of Beijing-You chicken (BJY). (**A**) The number of co-enrichment KEGG pathways of DEGs and SCMs. (**B**) Key pathways of flavor metabolites during BJY development. The blue or orange boxes below the metabolites indicate the contents during BJY development (from left to right: days 1, 56, 98, and 120). (**C**) Correlation between six hub DEGs and seven SCMs. Red indicates that there is a significant positive correlation between this gene and the metabolite, and green indicates a significant negative correlation (*p* < 0.05). The numbers indicate the corresponding correlation coefficient.

**Table 1 foods-12-01025-t001:** KEGG enrichment analysis of 82 intersecting genes.

Pathway	Pathway ID	Gene_ID	Differentially Expressed Genes
ECM–receptor interaction	gga04512	395210, 373987, 414837	*SPP1, THBS1, THBS2*
PPAR signaling pathway	gga03320	422345	*ACSL4*
Focal adhesion	gga04510	395210, 373987, 414837	*SPP1, THBS1, THBS2*
Melanogenesis	gga04916	408082, 395703	*EDNRB2, WNT11B*
Calcium signaling pathway	gga04020	408082, 395971, 428149	*EDNRB2, ADRB1, TBXA2R*
Fatty acid biosynthesis	gga00061	422345	*ACSL4*
Apelin signaling pathway	gga04371	395210	*SPP1*
Phagosome	gga04145	373987, 414837	*THBS1, THBS2*
DNA replication	gga03030	423688	*DNA2*
Fatty acid degradation	gga00071	422345	*ACSL4*

## Data Availability

Data that support the findings of this study are available upon request to the corresponding author.

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
