# Peer review of "Identification of Key Genes Affecting Flavor Formation in Beijing-You Chicken Meat by Transcriptome and Metabolome Analyses"

_foods, 2023, doi:10.3390/foods12051025_

Round 1

Reviewer 1 Report

The present study addresses transcriptome and metabolome analyses performed jointly to investigate the metabolic dynamics of flavour precursors in breast muscle samples from a local chicken breed. Apart from the technological parameters of meat quality, which are well-researched, the issue of flavour formation in meat - an issue of utmost importance to consumers - is less clear in terms of key regulatory mechanisms. The present study is thus very useful and up-to-date. Moreover, the study clearly benefits from the advanced analytical methods and the multidisciplinary approach that incorporates metabolomics data and the expression of selected genes related to the metabolism of meat flavour formation with age, which can provide new and important insights for a better understanding of these complex phenomena.

However, this is basic research and the prospects for selection to improve the taste of broiler meat based on the key metabolites and genes identified and discussed in this study need to be confirmed in further research.

Reviewer 2 Report

line 30 - 1st decissions of buying are based on meat color

line 31 - please add citations

lines 53-58 are not necessary and can be removed without harm to the manuscript; the reviewer would prefer to get more information about BIY chickens.

M&M 2.2 description of animals' keeping condition is too brief. Health conditions? Vaccinations? no lost birds during the experimental time? Any basic chemical analysis of meat compounds? Sensory analysis?

line 80- 10 individuals/time point? number of samples is limited

2.3 &2.5.1 the amount of samples used for RNA extraction& metabolite extraction?

Please improve figures& tables legends

line 82 - breast muscle with skin or without the skin?

Discussion is too general.

Limitations of this study?

Reviewer 3 Report

-         The manuscript foods-2111231 entitled; “Identification of Key Genes Affecting Flavor Formation in Beijing-You Chicken Meat by Transcriptome and Metabolome Analysis”. The authors studied the data of breast muscle at days 1, 56, 98, and 120 of Beijing-You chickens they found 310 significantly changed metabolites and identified 7,225 expressed genes. In my opinion, the article has good data and a good presentation. Also, the experimental design is clear enough.

-         General comments

-         Please proofread the whole manuscript to avoid grammatical errors.

-         Please describe all abbreviations in their first mention.

-         In the discussion section, please be more specific, discuss your study with other similar studies and please state the superiorities of your study when compared to previous ones.

-         In the references section, use journal style. 

Round 2

Reviewer 2 Report

Thank you for following my suggestions during revision of the manuscript.